# Cross Dataset Analysis for Generalizability of HRV-Based Stress Detection Models

**DOI:** 10.3390/s23041807

**Published:** 2023-02-06

**Authors:** Mouna Benchekroun, Pedro Elkind Velmovitsky, Dan Istrate, Vincent Zalc, Plinio Pelegrini Morita, Dominique Lenne

**Affiliations:** 1Biomechanics and Bioengineering Lab, University of Technology of Compiègne (UMR CNRS 7338), 60200 Compiègne, France; 2Heudiasyc Lab (Heuristics and Diagnosis of Complex Systems), University of Technology of Compiègne (UMR CNRS 7338), 60200 Compiègne, France; 3School of Public Health and Health Systems, University of Waterloo, Waterloo, ON N2L 3G1, Canada; 4Research Institute for Aging, University of Waterloo, Waterloo, ON N2J 0E2, Canada; 5Department of Systems Design Engineering, University of Waterloo, Waterloo, ON N2L 3G1, Canada; 6Institute of Health Policy, Management, and Evaluation, Dalla Lana School of Public Health, University of Toronto, Toronto, ON M5T 3M6, Canada; 7Centre for Digital Therapeutics, Techna Institute, University Health Network, Toronto, ON M5G 2C4, Canada

**Keywords:** heart rate variability (HRV), stress monitoring, e-health, IoT, stress bio-markers, generalizability

## Abstract

Stress is an increasingly prevalent mental health condition across the world. In Europe, for example, stress is considered one of the most common health problems, and over USD 300 billion are spent on stress treatments annually. Therefore, monitoring, identification and prevention of stress are of the utmost importance. While most stress monitoring is carried out through self-reporting, there are now several studies on stress detection from physiological signals using Artificial Intelligence algorithms. However, the generalizability of these models is only rarely discussed. The main goal of this work is to provide a monitoring proof-of-concept tool exploring the generalization capabilities of Heart Rate Variability-based machine learning models. To this end, two Machine Learning models are used, Logistic Regression and Random Forest to analyze and classify stress in two datasets differing in terms of protocol, stressors and recording devices. First, the models are evaluated using leave-one-subject-out cross-validation with train and test samples from the same dataset. Next, a cross-dataset validation of the models is performed, that is, leave-one-subject-out models trained on a Multi-modal Dataset for Real-time, Continuous Stress Detection from Physiological Signals dataset and validated using the University of Waterloo stress dataset. While both logistic regression and random forest models achieve good classification results in the independent dataset analysis, the random forest model demonstrates better generalization capabilities with a stable F1 score of 61%. This indicates that the random forest can be used to generalize HRV-based stress detection models, which can lead to better analyses in the mental health and medical research field through training and integrating different models.

## 1. Introduction

Stress is typically defined as the reaction people have when faced with demands and pressures bigger than their ability to handle [1], and it is an increasingly prevalent mental health condition across the world. While acute stress responses to situations are normal, constant long-standing stressors characterized as chronic can lead to severe health problems such as hypertension and cardiovascular diseases [1]. In Europe, for example, stress is considered one of the most common health problems, and it is estimated that over USD 300 billion are spent on stress treatments every year [2]. Therefore, the monitoring, identification and prevention of stress is of the utmost importance. Stress has usually been quantified through self-report metrics (e.g., questionnaires such as the Perceived Stress Scale [3]), which take into account the individuals’ subjective experience. This state, related to how individuals perceived threats in the world, is psychological stress. However, physiological parameters are linked to emotional states; for example, changes in heart rate, cortisol levels, blood volume pulse, brain activity and electrodermal activity are all linked with stress [1].

Typically, biomarkers and physiological data related to stress can be collected with the use of sensors in laboratory experiments, in which specific stressors can be applied to participants to induce acute stress [4]. However, for greater validity and applicability, stress-related data must also be collected and analyzed in real-world environments. In this context, allowing study participants to self-report their stress levels throughout the day might provide valuable insight into more chronic types of stress. The Ecological Momentary Assessment (EMA) methodology, which focuses on repeated measurements throughout the day to allow self-report closer to real experiences, can be used here [5]. In addition, many wearable technologies collect physiological data related to stress states. For example, heart rate variability (HRV)—the variation in time between consecutive heart beats and one of the most widely used stress-related features—can be collected with the use of wearable devices such as the Empatica E4 wristband.

Using self-reports and HRV data from wearable technologies allows researchers to access labelled data of stress states, in which the labels are the stress self-report of participants and the dataset is composed of physiological measurements. In this context, Machine Learning (ML) models can be used to predict the stress level of participants based on data collected in real-world environments. If successful, prediction models can go one step further than simply quantifying stress states and be used to identify, in real-time, if an individual is stressed and, in turn, prevention strategies can be used to reduce stress levels (e.g., a meditation app can be triggered).

Therefore, models based on real-life data can be developed in two ways: (i) a model is trained on more accurate and less noisy laboratory data; and tested on real-life, more representative data; (ii) a model is trained solely on real-life data.

The goal of this study was to develop proof-of-concept monitoring models built on data collected in daily life environments using these two approaches and focusing on HRV data. First, a model was trained based solely on data collected from a laboratory experiment, and a model based solely on data collected in daily life. Then, laboratory data were used to train a model which was tested on real-life data, in order to validate the generalizability of the models.

The paper is divided as follows: after a brief related work overview is presented, Section 2 describes materials and methods, including experimental protocols for the two studies (laboratory and daily life), as well as the ML models used in the analyses. Section 3 presents the results and discussion, and Section 4 finishes the paper with our final conclusions.

### 1.1. Related Work

Many studies [1,6,7,8] provide a review of the features and AI methods used in stress prediction as well as a survey of stress prediction in laboratory and real-life scenarios with mobile and wearable technologies. Among the many details presented, of note HRV is widely used for stress prediction. In addition, the accuracy for methods that use HRV in laboratory environments lies somewhere between 70–90%, and the accuracy tends to fall in unrestricted daily life experiments to somewhere between 60–80%. Daily life experiments also tend to use a suite of behavioural and physiological parameters.

A study carried by Can et al. [4] defines a nomenclature for four types of stress detection studies: laboratory-to-laboratory known context (LLKC), laboratory-to-laboratory self-report (LLSR), daily-to-daily self-report (DDSR), and laboratory-to-daily self-report (LDSR). Using this nomenclature, our study will focus on DDSR and LDSR types. In their study, the authors collected self-reports and physiological measures with the Empatica E4 in a laboratory experiment, and also collect daily life data using the Empatica and an EMA methodology for the self-report, collecting four data points over a period of seven days. Using only real-life HRV data, they achieved results from 65% to 68.30% across several models with 10-fold cross validation, with the best models being a Multi-layer Perceptron and Logistic Regression (LR). When using laboratory data combined with real-life self-report, the accuracy increased to over 70% for most models, with best results being observed with the Random Forest (RF) and Logistic Regression models (accuracy of 71.78%) and the Support Vector Machine (SVM, 73.44%).

Giannakakis et al. [9] investigated HRV parameters in order to recognize stress using different machine learning algorithms. They used a personalized baseline for each participant in order to eliminate inter-subject variability and use it to normalize data. Models were applied to HRV features before and after a pairwise transformation. The selected features fed machine learning systems achieving a classification accuracy of 84.4% for the transformed features with a Support Vector Machine (SVM) model and 75% accuracy with a RF model without transformation.

R. Castaldo et al. [10] also used different models to detect stress from linear and non-linear HRV features. The dataset they used was recorded from 42 university students, during oral examination (stress) and at rest after a vacation. They reached around 80% accuracy with all the models including the SVM, C4.5 and AdaboostM1.

Hovsepian et al. [11] trained an SVM using Electrocardiograms (ECG) and respiration data in both laboratory (n = 24) and real-life settings (n = 20, 1-week data). The model predicted the probability that a user was stressed with an accuracy of 90% in the laboratory and 72% in real-life. Gjoreski et al. [12] combined SVMs and RFs, using a wrist device to collect acceleration and physiological measures in laboratory and real-life. The data from the laboratory was used to build a stress detector on 4-min windows with RFs showing a 83% accuracy score. SVMs were trained with the last 10 outputs of the RF detector and an activity-recognition algorithm to classify 20 min of data according to the stress level detected. The accuracy without this additional context is 76%, increasing to 92% with the SVM detector.

It is true that stress detection using HRV features have been investigated thoroughly in various studies, some of which are cited above. Several AI models are used on different datasets and many of them yield fairly high accuracy scores. However, the comparability of these studies is very difficult given the many differences in terms of sensors, stressors and preprocessing approaches. Furthermore, the generalizability of these models is almost never discussed.

One way to tackle this problem would be to use cross dataset analysis to study the generalizability of a model and measure of how useful the results of a study are for a broader application. Although there is intensive research in the field of stress monitoring from physiological signals, there is only a few publicly available datasets.

One of the first stress datasets was published by Healey et al. [13]. It included ECG, trapezius muscle signal, skin conductance and respiration from 24 subjects during 50 min real-world driving. Protocol validation was achieved by administering questionnaires and conducting video coding for some of the subjects. This is one of the rarer studies to have been carried out in an ambulatory environment.

Another stress dataset of note is the SWELL dataset [14]; composed of a variety of data such as HRV, body postures, facial expressions, and skin conductance, providing interesting behavioural data in addition to physiological ones. The dataset was collected from an experiment of 25 subjects in their work spaces while they performed typical knowledge work under stressful conditions such as time pressure and email interruptions. Various data including computer logging, facial expression, body postures, ECG and skin conductance were recorded in neutral and stressful conditions.

The Database for Emotion Analysis using Physiological Signals (DEAP) [15] is composed of Electroencephalograms (EEG) readings and other peripheral physiological signals such as Galvanic Skin Reponse (GSR), Electromyograms (EMG), respiration among others, from 32 participants watching music videos together with self-report.

Furthermore, Schmidt et al. [16] recently used the Wearable Stress and Affect Detection Data Set (WESAD) which uses the Empatica E4 as well as a chest-band to collect respiration data, with self-reports composed of several stress questionnaires for ground truth. Using HRV features in addition to Electrodermal Activity (EDA), temperature and others, the authors achieved accuracies on a variety of ranges from around 60% to 90% depending on the data used and the model, for binary stress classification. Of note for our work, the RF model achieved an accuracy of 88% for Empatica data.

Soleymani et al. [17] created the MAHNOB-HCI multimodal database. They recorded the peripheral physiological signals including ECG, GSR, Skin Temperature and respiration from 24 participants after eliciting their emotion by 20 affective movies.

### 1.2. Paper Contribution

This paper presents one of the first explorations of the generalizability of HRV-based machine learning models for stress detection. The study uses a specific approach, including a leave-one-subject-out cross-validation and a cross-dataset analysis, which are more representative of real-world applications. Furthermore, training the models on laboratory data and testing them on real-world data makes the study particularly noteworthy. Laboratory data are often more controlled and consistent than real-life data, making it easier to obtain accurate measurements and labels. Additionally, it is typically collected in a controlled environment, which can reduce the amount of noise and variability in the data. On the other hand, real-world data often contain more variability and noise, making it a more challenging test for the model. Evaluating the model on ambulatory data can help identify any limitations or weaknesses in the model and provide insights into how it can be improved. Moreover, multiple evaluation scores are computed in order to have a comprehensive understanding of the model’s performance. They provide different insights into the model’s performance and can help identify limitations and allow to take into account the specific costs of different types of errors. This research makes a significant contribution to the field by showing the applicability of HRV-based models for stress detection in real-world settings. It highlights the potential of these models to be widely used in various populations and environments.

## 2. Materials and Methods

### 2.1. Datasets

In this study, two different datasets were used to test ML model’s generalizability on data collected in different setups using different sensors. The experimental studies were conducted independently on different subjects in distinct time frames. They were approved by unrelated institutional review boards and IRB numbers were obtained for both datasets introduced below.

#### 2.1.1. Multi-Modal Stress Dataset (MMSD) for Real-Time, Continuous Stress Detection

MMSD [18] is a multimodal acute stress dataset from a laboratory study (IRB 202077, 31 August 2020). It includes physiological signals such as Electrocardiograms (ECG), Photoplethysmograms (PPG), Electrodermal Activity (EDA) and Electromyograms (EMG) from 74 subjects recorded in three different states successively: 15 min relaxation; 20 min stress and 10 min recovery. The sample includes 38 women aged 19 to 63 years old (mean age: 33 y.o ± 12.5) and 36 men aged 21 to 79 years old (mean age: 35 y.o ± 13). A representative sample of the French population in terms of age and gender was chosen for the study. Only healthy volunteers who did not have any chronic diseases such as chronic stress, cardiovascular diseases or other conditions that could affect the collected signals were included [18].

The initial phase of the study, in which participants were in a state of relaxation, was used as the benchmark. A measurement of cortisol taken immediately after this phase was used to compare to a second measurement taken after a stress phase, in order to confirm the effectiveness of the stressors used in the study. Stress was induced using serious games such as Stroop Color Word Test (SCWT) and mental arithmetic. The validity of the protocol was established using both psychological questionnaires and a physiological measure of cortisol before and after the stress phase. The State-Trait Anxiety Inventory form S (STAI-S) was used as a self-report of state anxiety. It consists of 20 items that ask subjects to rate how they feel at the present moment. It is commonly used in research and clinical settings to assess stress levels in adults [19]. Since only volunteers without chronic stress were included in the study, there was no need to use a baseline measurement for chronic stress. The increase in both cortisol levels and STAI-S scores after exposure to the stressors was sufficient to confirm the presence of acute stress. In this paper, only HRV signals obtained from ECG signals were analyzed. The first phase of the study, in which participants were in a state of relaxation, and the third phase, in which they were in a state of recovery, were combined into one category as a non-stress or baseline condition because both are characterized by a reduction in the body’s physiological arousal. During relaxation and recovery, the body’s parasympathetic nervous system is activated, which promotes relaxation and rest whereas during a stress state, the body’s sympathetic nervous system is activated, causing the release of stress hormones and other physiological responses in response to a perceived threat.

#### 2.1.2. University of Waterloo Stress Dataset (UWS)

UWS dataset [20] contains physiological data including HRV, EDA, and ECG from 27 subjects recorded during two weeks in their personal, real-life environment. The sample includes 19 women—32% aged 18–24; 27% aged 25–34; 26% aged 35–44; 5% aged above 65%—and 8 men—25% aged 18–24 and 25–34; 50% aged 35–44. Participants were recruited from a random sample in Ontario, and due to difficulty in finding participants unhealthy individuals (i.e., answered yes to having chronic disease, illness, smoking or drug use) were allowed (for this sample, 5 women responded yes to the aforementioned question). Since this dataset is comprised of real-life data, it is important to note no control or baseline was used.

The Empatica E4 wristband is used to record data and the main objective was to identify stress in ambulatory, daily life conditions. Labelling is based on self-reports collected approximately every three hours through the Depression, Anxiety and Stress Scale (DASS-21) using only questions related to stress. It consists of 7 items, each rated on a four-point Likert scale (0–3) based on how much the item has applied to the person over the past three hours. Data segments are labeled as stress when the DASS questionnaire following the segment indicates a score higher than 7, which on DASS-21 represent mild levels of stress and above [21]. Subjects reported being stressed 17% of the time on average. Overall, 70% of the UWS dataset was comprised of female participants. In this paper, HRV signals recorded by the Empatica E4 are analyzed for stress identification. This dataset collection was approved by the University of Waterloo Research Ethics Board (IRB 43612, 6 October 2021).

### 2.2. Heart Rate Variability Analysis and Feature Extraction

HRV is a measure of the RR intervals and is often used as a measure of stress because it reflects the balance between the sympathetic (SNS) and parasympathetic (PNS) branches of the autonomic nervous system. The RR interval, or “normal-to-normal” (NN) interval, is the time interval between consecutive heartbeats in an ECG signal. The normal range for RR intervals may vary depending on the individual and their physical condition [22,23]. The typical range is between 300 and 1300 ms, or 0.3 and 1.3 s. This equates to a heart rate of 45 to 200 beats per minute [24]. When the body is in a state of stress, the SNS is more active leading to a less variable heart rate and thus a decrease in HRV. Conversely, when the body is in a state of relaxation, the PNS is dominant leading to an increase in HRV [25]. Therefore, measuring HRV can provide insights into an individual’s stress levels through several features. In this study, HRV analysis is performed using various Python Toolboxes including HRV [26], Time Series Feature Extraction Library (TSFEL) [27], Python toolbox for Heart Rate Variability (pyhrv) [28]. Features are computed over side-by-side 5 min segments for the UWS dataset. On the other hand, for the MMSD dataset, 5 min segments with a 1 min sliding window is used to increase the number of analyzed windows.

Pre-processing steps are required prior to any HRV analysis in order to handle data losses and bias caused by errors in the signal.

They include:Deletion of abnormal beats (RR>1.3 s or RR<0.3 s)Data imputation

Since missing data affect HRV features and may lead to false conclusions, a quality criteria is used to discard segments with a high percentage of missing data ($P_{missing}\leq 50\%$). It is computed over each window as explained in the pseudo code below (Algorithm 1).
**Algorithm 1** Percentage of missing data for each HRV window1:**for** each RR[i] in the window (i from 1 to length of HRV window) **do**2:    Compute         gap=T[i]−T[i−1]3:    **if** gap> 1.3 s **then**4:        Compute mean RR over last 10 values ($mean_{RR}$)5:        Compute *N*, the number of missing RR in the gap:         $N=\mathrm{Floor}\mathrm{value}\;\;(\frac{gap}{RR_{mean}}$)6:    **end if**7:    Compute $N_{tot}$, the total number of missing RR in the window $N_{tot}=\sum N$8:    Compute percentage of missing data: $P_{missing}=100*\frac{N_{tot}}{\mathrm{Length}\mathrm{of}\mathrm{window}+N_{tot}}$9:**end for** RR[i] is the RR value at position i, and T[i] its timestamp

Only 5 min segments that met the quality criteria are interpolated using a shape-preserving piecewise cubic Hermite interpolating polynomial (PCHIP) before HRV analysis [29].

The missing data threshold ($P_{missing}\leq 50\%$) and imputation approach (PCHIP interpolation) were selected based on the findings of a previous study [29]. There is likely to be an error in the estimation of HRV parameters due to the interpolation of missing data. The estimation error can vary depending on the specific parameter, some HRV features being more sensitive to interpolation than others [29]. Moreover, deleting abnormal heartbeats in HRV signals can lead to data loss. While ectopic beats often occur due to false peak detection, they can also be caused by physiological phenomena such as premature ventricular or atrial contractions [30]. In these cases, suppressing ectopic peaks may remove important physiological information from the analysis. It is therefore important to consider the potential impact of ectopic peak suppression on the HRV analysis.

A total of 61 features in the time, frequency, and non-linear domains are computed among the most used HRV features, and are listed in the Table 1 along with the Python toolbox used for each set of features.

In this study, frequency domain analysis is performed using Fast Fourier Transform (FFT) on signals re-sampled at 8 Hz. HRV spectrum is aggregated into very low frequency (VLF) (0.003–0.04 Hz), low frequency (LF) (0.04–0.15 Hz), and high frequency (HF) (0.15–0.4 Hz) [31].

**Table 1 sensors-23-01807-t001:** List of computed HRV features.

Toolbox	Time Domain	Frequency Domain	Non Linear
pyHRV [32]	Number of NNi, Mean NNi, Maximum NNi, Mean heart rate, Min, Max heart rate, Heart rate standard deviation, Mean NNI difference, Min, Max NNI difference, sdnn, rmssd, sdsd, nn50, pnn50, nn20, pnn20, $tinn_{n}$, $tinn_{m}$, tinn, tri_index	Total power, Very low frequency, Low frequency, High frequency, $\frac{\mathrm{Low}\mathrm{frequency}}{\mathrm{High}\mathrm{frequency}}$ Normalized low frequency, Normalized high frequency	SD1, SD2, $\frac{\mathrm{SD}1}{\mathrm{SD}2}$
TSFEL [27]	Absolute energy, Auto correlation, Median, std, Entropy, Inter-quartile range, Mean absolute deviation, Mean absolute difference, Mean, Median difference, Median absolute deviation, Median absolute difference, Kurtosis, Skewness, slope	Band-width, Spectral entropy, Spectral decrease, Spectral distribution, Spectral kurtosis, Spectral roll 95, Spectral skewness, Spectral variance, Spectral centroid, Median frequency Energy, Human energy Fundamental frequency Area under the curve of the signal	

Table 2 summarizes the datasets dimensions with class sizes and number of subjects.

The dimensions of the UWS dataset are much greater than the MMSD dataset despite the number of subjects being lower. This is because in the first dataset, subjects were recorded every day for two weeks compared to the 40 min duration for each subject in the latter dataset. In addition, as can be seen from Table 2, since the data were collected in real-life without any stress stimuli, the UWS dataset is extremely imbalanced with a positive class (stress) representing less than 7% of the entire dataset. This implies that specific pre-processing methods adapted to imbalanced datasets should be used prior to any analysis.

### 2.3. Classification Models

A logistic regression model (LR) was used in this study. This model is usually relevant when the objective is to predict a categorical dependent variable—such as in this case Stress (2) VS No Stress (1). The ’LogisticRegression’ package from the Sklearn library in Python was used.

A Random Forest model (RF) was also used to classify stress versus non-stress states in both datasets due to its numerous advantages, such as robustness to outliers and non-linear data and requiring very little pre-processing. Moreover, over-fitting can be prevented thanks to simple hyper-parameters optimization and pre-pruning techniques by prematurely stopping the growth of the decision tree. The model was implemented using the ‘RandomForest’ package from the Sklearn library in Python.

A grid search with a 10-fold cross-validation was used for hyper-parameter optimisation. The model was optimized over the ROC AUC score since in this case, positive and negative classes are equally important. Hyper-parameters tuned to prevent the model from over-fitting include min-samples split, the minimum number of samples required to split an internal node and min-samples-leaf the minimum number of samples per leaf node The number of decision trees grown based on a bootstrap sample of the observations n-estimators and the number of features to consider when looking for the best split max features were also tuned in the grid search.

The grid search was only performed on the MMSD dataset and the same hyper-parameters are used in all analyses. This choice is due to the quality of data, which is assumed to be better in laboratory conditions and the fact that labels are more accurate.

The analysis’ steps were two-fold:Independent analysis: the RF and LR models are used to analyze MMSD and UWS datasets independently using a Leave-one-subject-out (LOSO) approach.Cross dataset analysis: The RF and LR models are trained on MMSD dataset and tested on the UWS. The end goal is to assess the models’ ability to adapt properly to new, previously unseen data.

Specific pre-processing steps are required before using any ML model to avoid false conclusions caused by extremely imbalanced datasets.

First, extreme outliers are removed using the 1.5 Interquartile Range (IQR) method [33].
(1)$$\begin{array}{c} \mathrm{IQR}=Q3-Q1. \end{array}$$
where Q1 is the first quartile of the data, i.e., 25% of the data lies between minimum and Q1. Q3 is the third quartile of the data, i.e., 75% of the data lies between minimum and Q3.

To detect the outliers using this method, a new range was defined and any data point lying outside this range is considered as an outlier and is deleted. The range is as given below:(2)$$\begin{array}{c} \mathrm{Lower}\mathrm{bound}:(Q1-1.5*\mathrm{IQR}). \end{array}$$(3)$$\begin{array}{c} \mathrm{Upper}\mathrm{Bound}:(Q3+1.5*\mathrm{IQR}). \end{array}$$

When scale is taken as 1.5, then according to the IQR method any data which lie beyond 2.7σ from the mean (μ), on either side, shall be considered as outlier. This decision range is the closest to what Gaussian Distribution considers for outlier detection.

Secondly, in order to balance the classes, the training set is upsampled using the SMOTE [34] method in order to avoid over-fitting over the negative class. Only the train set is upsampled to make sure the model is tested on original real-life data. Finally, both train and test sets are scaled using the RobustScaler from Sklearn library. This Scaler removes the median and scales the data according to the IQR range computed on the samples in the training set. Relevant statistics are then used to transform the test set.

The split was achieved over subjects so that windows from the same subject would always be in the same set. This is in order to avoid contamination between train and test sets. A (LOSO) cross validation was used for an accurate estimate of model’s performance and because of the small size of the datasets. This method is also more representative of reality where the end goal would be to classify data from each new subject.

### 2.4. Evaluation Metrics

A number of metrics were used in order to have a deep understanding of our model’s performance especially given the extremely imbalanced datasets. F1-score is computed as the harmonic mean of precision and recall where:
(4)$$\begin{array}{c} \mathrm{Precision}=\frac{\mathrm{TP}}{\mathrm{TP}+\mathrm{FP}},\\ \mathrm{Recall}=\frac{\mathrm{TP}}{\mathrm{TP}+\mathrm{FN}}, \end{array}\;\;\;\;\;\;\;\;\;\;\;\;\;\;\;\;⟹F1=\frac{2*\mathrm{Precision}*\mathrm{Recall}}{\mathrm{Precision}+\mathrm{Recall}}.$$
where

TP = True Positive, HRV windows of stress classified as stress,

FP = False Positive, HRV windows of relaxation classified as stress,

FN = False Negative, HRV windows of stress classified as relaxation.

To tackle the class imbalances, the macro-average f1-score is computed given the fact that classes are equally important.

Metrics are also presented for each class in order to see how well each class is recognized by the RF model.

Moreover, ROC AUC scores are computed since they capture more than one aspect of the classification by taking both false positives and negatives into account through the True Positive Rate (TPR) and False Positive Rate (FPR):


(5)
$$\begin{array}{c} \mathrm{TPR}=\frac{\mathrm{TP}}{\mathrm{TP}+\mathrm{FN}},\\ \mathrm{FPR}=\frac{\mathrm{FP}}{\mathrm{FP}+\mathrm{TN}}, \end{array}\;\;\;\;\;\;\;\;\;\;\;\;\;\;\;\;⟹\mathrm{AUC}\mathrm{ROC}=\frac{1+\mathrm{TPR}-\mathrm{FPR}}{2}.$$


Finally, specificity, also known as true negative rate was computed. It is a measure of the model’s ability to correctly identify negative examples (non stress).
(6)$$\begin{array}{c} \mathrm{Specificity}\;\left(\mathrm{TNR}\right)=\frac{\mathrm{TN}}{\mathrm{TN}+\mathrm{FP}}, \end{array}$$

This performance metric is relevant in the context of imbalanced datasets because the model may be trained to prioritize accuracy over identifying the negative examples of the minority class. By using specificity as a performance metric, it ensures that the model is able to identify the majority class (non stress) with a high degree of accuracy, while still maintaining a high sensitivity for the minority class (stress).

Python functions were used to compute the metrics namely “f1_score” and “roc_auc_score” from Sklearn Library. Scores from the cross-dataset analysis are an indicator of the generalizability of the model trained with laboratory data on signals collected in an ambulatory environment. By using the (LOSO) cross-validation, it is possible to avoid both over-fitting and train/test sets contamination.

## 3. Results and Discussion

In this section, results from our evaluations are presented for each analysis. As explained above, the RF model parameters are chosen using a grid search on the MMSD dataset. The grid search yielded the hyper-parameters summarized in Table 3.

This model is used to classify stress versus non stress conditions in all analyses introduced above using a (LOSO) cross validation for MMSD and UWS analyses.

### 3.1. Independent Analysis

Results from the independent datasets are presented in Table 4 and Table 5, best scores are highlighted in bold.

The LR model outperformed the RF model on the MMSD dataset. Using the LR model, both Stress and Non Stress classes are well recognized with an f1 score of 71% for each class and an accuracy of 74%. For the UWS dataset on the other hand, the RF model gave better results but the stress class is less well recognized. Although good classification scores are reached on the UWS dataset, it is important to note that one questionnaire is used to label the previous three hours, and thus, these results may be biased due to weak labelisation. Weakly labeled data can pose challenges for the models because they may not have enough information to make accurate predictions.

Table 6 summarizes some of the studies that have used RF and LR models for stress detection as well as the accuracy obtained in the datasets used in this study. Both models achieve similar results using only HRV features, without combining multiple modalities, as opposed to some of the studies in the table. It shows that the RF model achieved a slightly lower accuracy of 67% on the MMSD dataset, however, it had a higher accuracy of 81% on the UWS dataset.

This discrepancy in performance can be attributed to various reasons such as the dataset size and diversity. Models perform best when the data are diverse and contain a variety of features and samples. Furthermore, the use of different physiological signals in some studies may also account for the good results. This is because stress can manifest in differently in various physiological systems, and by considering multiple signals, the model may be able to better capture these different stress responses.

Even using only HRV features, some studies have achieved slightly better scores using RF and LR models compared to the results obtained in this study [36,37,38]. However, the use of the Leave-One-Subject-Out (LOSO) cross-validation method in this study may account for the lower performance. An RF model applied the MMSD dataset without using a LOSO cross-validation, resulted in an f1 score greater than 80% [39]. Studies using the same procedure also tend to yield similar results [40,41]. The main advantage of LOSO is that it makes the most efficient use of the available data. It provides an estimate of the model’s performance on new data, which is important when the goal is to generalize the model to new subjects. Additionally, LOSO is especially useful when the number of subjects is small as is the case for the datasets used in the study.

It is also important to keep in mind that accuracy, as presented in Table 6, is not an effective metric for evaluating imbalanced datasets. In such cases, the majority class can skew the accuracy metric, leading to a false sense of high model performance even if the model is not accurately predicting the minority class. Other metrics such as f1-score and class-specific metrics are more appropriate for evaluating performance on imbalanced datasets. For instance, even though the LR model on the UWS dataset has a seemingly high overall accuracy of 70%, it still has poor performance on the minority class (stress) with an f1-score of only 45%.

Next, RF and LR models are evaluated using cross-database validation as a semi-supervised learning framework where models are trained on the MMSD dataset and tested on UWS dataset. The results of cross-dataset evaluations are presented in Table 7.

### 3.2. Cross Dataset Analysis

The same f1 score of 63% is reached in the cross-dataset analysis and the MMSD independent analysis using the RF model. For the LR model on the other hand, a loss of about 14% can be noticed on the f1-score in the cross-dataset analysis. The stress class seems to be less well recognized by both models. Although the LR model gave better classification results in the independent analysis of the MMSD dataset, the RF model is more generalizable.

This is because the RF model is less prone to over-fitting issues thanks to parameters tuning. Moreover, for the final decision, the RF classifier aggregates the decisions of individual trees; consequently, RF classifier exhibits good generalization capabilities.

In [41] as well, the RF model was found to have good generalizability across datasets. The results showed that while RF achieved slightly lower F1-score and Accuracy than other HRV based models on within dataset evaluations for binary stress classification, it outperformed them in the cross-dataset analysis. Moreover, ref. [42] also demonstrates the generalization capabilities of RF models across different datasets using HRV features alone and coupled with electrodermal activity features. The results from both studies confirm that RF is robust against variations in environmental factors such as recording devices or types of stressful scenarios.

According to ROC-AUC scores in Table 7, RF and LR models equally identify true positive cases (stress). The LR model however has a higher specificity which indicates that the model is able to correctly identify more true negative cases (non stress). Both models are considered to have an average performance. A higher AUC-ROC and specificity is desired but a balance between these two metrics needs to be achieved.

One of the main challenges faced in the study is the inter-individual and measurement variability. Since HRV can vary greatly between individuals and can be affected by various factors such as measurement equipment, recording conditions, and physiological factors, it is difficult to develop a model that is applicable to a wide population. In this study, labels in the UWS dataset may not be as accurate, and the stress response may vary between the two datasets as the MMSD dataset uses artificial stressors whereas the UWS dataset uses stressors from daily life events. Additionally, the devices used to collect data in the two datasets are also different, which could affect the generalization of the models and contribute to the lower stress class recognition scores.

In the MMSD dataset, the inter-individual variations were taken into consideration in the experimental protocol by controlling the stressors, the environment and subject-related variations by establishing a subject profile [18]. By training the models on laboratory data (MMSD dataset), it is possible to ensure that they are able to accurately identify the relevant features or patterns associated with stress. On the other hand, testing on ambulatory data (UWS dataset) allows for an evaluation of its performance in real-world scenarios, and can help identify any limitations or challenges that need to be addressed in order for the model to be effectively used in these settings. Overall, training on laboratory data and testing on ambulatory data is a way to evaluate the model’s performance on different types of data and ensure that it is robust and generalizable enough to be used in real-world settings.

Furthermore, the lack of studies on the subject of model generalizability for stress detection makes it difficult to compare results across studies and fully support the claims of our study. Model generalizability is a crucial aspect of any machine learning study, as it determines how well the model will perform on new, unseen data. However, in the field of stress detection through HRV signals, there is only little research on how well models generalize to new subjects and conditions. This lack of studies makes it challenging to draw strong conclusions about the generalizability of our own model and its performance on new data. Furthermore, the lack of a standardized evaluation protocol for stress detection models also makes it challenging to compare the performance of our model to other models in the literature.

A standardized protocol together with data analytics can play a significant role in the understanding of individual’s stress response and management, by providing insights and predictions about individual’s stress levels. This would pave the way for early intervention, and evaluation of the effectiveness of different stress management techniques. Furthermore, it can potentially contribute to medical conditions’ prognosis by using the predictions of stress levels as a marker for other health conditions related to stress. By developing a more accurate and generalizable stress diagnostic and monitoring model, it will be possible to better identify and manage stress in individuals. This can lead to improved health outcomes and potentially reduced healthcare costs. Additionally, such a model could be used in a variety of settings, such as workplaces and schools, to help promote mental health and well-being. It is important to note that these claims need to be further validated through rigorous testing, and the generalization capability of the model needs to be evaluated on different population and demographic groups.

Furthermore, it is important to keep in mind that stress is not a binary problem. It is a continuous variable that can have different levels of intensity and duration. Low levels of stress can be beneficial as they can improve motivation and performance, but high levels of stress can have negative effects on physical and mental health. Thus, stress should actually be considered as a multi-class problem where the model should be able to classify levels of stress according to a rating scale. It is however a more complex problem and binary stress detection can be considered as the first step towards multi-class stress classification. Once the binary stress detection model is developed and validated, it can be used as a pre-processing step to filter out instances that are not stress. A second model can then be trained on a dataset that includes labelled instances of different levels of stress according to a standardized scale such as the DASS21 used in the UWS dataset. The scores can be interpreted based on the cut off points provided by the creators of the scale, which are: Normal: 0–7, Mild stress: 8–9 Moderate stress: 10–12 and Severe stress: 13–16. Such models can be trained to recognize patterns in the HRV data that are specific to different levels of stress.

## 4. Conclusions

As the Internet of Things (IoT) becomes more prevalent, there is growing interest in utilizing HRV as a means of remotely and continuously monitoring stress in a non-invasive manner. However, a major challenge in this field is the availability of data for stress recognition. This, combined with the ethical considerations surrounding the collection of biomedical data, highlights the need for models that are able to accurately predict stress across various datasets and populations.

This study aims to evaluate the generalizability of two HRV-based models, logistic regression (LR) and random forest (RF), by using two stress datasets (MMSD and UWS) differing in many aspects including stressors’ type, intensity and acquisition devices. The study first conducts an independent analysis on each dataset using a LOSO cross-validation, followed by a cross-dataset analysis. The use of laboratory data from the MMSD dataset in the training set ensures that the models identify only the relevant features and patterns associated with stress. The models’ performance is then evaluated on ambulatory data to ensure that they are robust and generalizable enough for real-world applications. Fairly good classification scores are obtained in the independent and cross-dataset analysis with better generalisation capabilities demonstrated by the RF model (62% F1 score), although it performed slightly less well than the LR model on MMSD independent evaluation. These results align well with those obtained in a similar study found in literature using the same standards (LOSO and cross-dataset analysis) [41].

Future work could expand upon this study by testing the models on additional datasets, as well as using other machine learning models. Specifically, deep learning models, such as Convolutional Neural Networks (CNNs), with HRV time-series data could potentially lead to more accurate classification results. Additionally, it would be valuable to understand the specific factors that impact the generalizability of the models, such as the type of acquisition device used, the type and intensity of stressors and other features. This could be achieved by controlling for certain factors and comparing the models’ generalizability across datasets.

Studies on the generalizability of models using various physiological signals for ambulatory stress detection could have several potential benefits on Point-of-care testing (POCT), where stress can be monitored at patient’s location. This can be particularly beneficial for patients who have mobility issues. It can also improve early detection and intervention of stress and better management of chronic conditions affected by stress such as hypertension, diabetes and heart diseases.

## Figures and Tables

**Table 2 sensors-23-01807-t002:** Number of subjects and HRV windows for each dataset.

Dataset	Subjects	Positive Class (Stress)	Negative Class (Non Stress)	Total Size
MMSD	68	622	818	1440 × 60
UWS	25	532	7560	8092 × 60

**Table 3 sensors-23-01807-t003:** Grid Search parameters for the RF model.

RF Hyper-Parameters
criterion = ’entropy’, max_features = ’sqrt’, max_depth= 6 min,_samples_split = 2, n_estimators = 200

**Table 4 sensors-23-01807-t004:** Classification scores from MMSD dataset using RF and LR models.

Model	F1	Specificity	ROC AUC	F1 Stress	F1 Non Stress
RF	0.63	0.54	0.7	0.58	0.68
LR	**0.71**	**0.66**	**0.76**	**0.71**	**0.71**

**Table 5 sensors-23-01807-t005:** Classification scores from UWS dataset using RF and LR models.

Dataset	F1	Specificity	ROC AUC	F1 Stress	F1 Non Stress
RF	**0.75**	**0.7**	**0.79**	**0.63**	**0.86**
LR	0.61	0.69	0.7	0.45	0.76

**Table 6 sensors-23-01807-t006:** Summary of ML models used for stress detection through physiological signals.

Study	Markers	Accuracy
Can et al., 2020 [4]	EDA, HRV, HR	74.6% (RF) 73.8% (LR)
Muaremi et al., 2014 [35]	ECG, HRV, Respiration, Temperature, skin response, Posture, Accelerometer, Sleep	71% (RF) 52% (LR)
Hovsepian et al., 2015 [11]	ECG, Respiration	72% (Support Vector Machine)
Giannakakis et al., 2019 [9]	HRV	84.4% (Support Vector Machine) 75% (RF)
Castaldo et al., 2016 [10]	HRV	80% (Decision Tree)
Schmidt et al., 2018 [16]	ECG, EDA, EMG, accelerometer, respiration, temperature, HRV	88% (RF)
MMSD [18]	HRV	67% (RF) 74% (LR)
UWS	HRV	81% (RF) 70% (LR)

**Table 7 sensors-23-01807-t007:** Classification scores cross dataset analysis.

Model	Macro Avg Recall	Macro Avg Precision	F1	Specificity	ROC AUC	F1 Stress	F1 Non Stress
RF	0.62	0.63	0.63	0.41	0.65	0.41	0.85
LR	0.59	0.61	0.57	0.62	0.64	0.39	0.70

## Data Availability

Part of the data presented in this study will be made openly available shortly.

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
