# Peer review of "Cross Dataset Analysis for Generalizability of HRV-Based Stress Detection Models"

_sensors, 2023, doi:10.3390/s23041807_

Round 1

Reviewer 1 Report

Writing issue:

- Rewrite the abstract to be more concise and avoid using any reference on it

- Give the fullname before its abbreviation. For example: No any fullname given for "AI" in the abstract.

Technical issue:

-  Unbalanced UWS dataset between "stress" and "non-stress", how to deal with this?

- You might calculate the TN and evaluate the specificity as well.

- The results are not that precise. You can consider to use convolutional neural network for the classification as well, that might improve the results.

- Please have a comparative table with previously conducted studies in discussion part?

Author Response

Thank you very much for your time and thoughtful corrections. All comments were addressed in the paper and referenced in the response letter here attached.

Reviewer 2 Report

Dear Authors,

The manuscript presents one practical approach, the authors' follow-up work to AI-based stress monitoring, which has real potential. It is well-written.

There are some needs for improvement to increase the impact of the research and help the readers have a clear big-picture from ML and healthcare application points of view. Kindly consider the following inputs, mainly from the healthcare and paper writing perspectives:

General:

(a) Present the terminology in full and then use the acronyms in the text, while the titles and subtitles better comprise the terminology in full instead of abbreviations.

(b) Edit and phrase attentively: typo, past tense, neutral voice rather than personal (e.g., it, the study, instead of we)

Abstract: Organise it to support the motivation and claims: e.g., the current worldwide medical and financial impact of stress and the current technology to monitor it (there is no need for citations here), the claims (if the study intends to be a diagnostic and monitoring tool since the sensitivity was calculated), the working principle investigated (originality, is it a proof of concept?, is it POCT?), the conclusion based on the proposed model's with a better generalisation capability (how the results may benefit the medical investigations panel and mental healthcare).

Introduction: a progressive flow will help readers understand the motivation and the proposed solution to the problem: e.g., the normal body state, the response to stressors as acute and chronic stress, the problem (higher incidence of stress not so well diagnosed and monitored with the current measurement methods), the challenge (new approaches to involve AI and IoT as POCT for personalised medicine), the solution (the paper to present the model for better generalisation), the impact and possible implementation (at POCT level, as wearables and personalised medicine). Citations of the authors' previous work will highlight the value of the present work.

(a) Present the concepts the study is focused on, with evidence from the scientific literature. e.g., define stress from physiological and psychological perspectives to support the motivation with specific data about the current situation worldwide (prevalence and incidence worldwide, medical and financial burden); briefly describe stressors and their acute and chronic impact on health to support the choice of HRV and cortisol; briefly compare acute and chronic stress impact and link to the claim of the study; formulate the motivation from a stress point of view and link to the measurement and monitoring tools; AI and the impact to medical investigations, with examples of analytical tools to support the claim and the novelty; POCT, IoT and wearables; the need of generalisation and the impact on the stress monitoring devices to highlight the claims.

(b) Briefly explain the prediction as one factor essential to support the development of new methods to detect and monitor stress from the point of view of the current methods and tools (comparison between the methods).

(c) Integrate/ blend the 1.1. Related work within the Introduction to support the advances in stress measurement procedures with advantages and disadvantages in terms of methods, sensitivity, specificity, challenges, and stress the paper’s claims once more.

(d) Conclude the Introduction with a paragraph that describes the entire study like an abstract (page 2, paragraphs 2-3: "... The goal of the study … on HRV data.") to highlight the claims and the potential again.

Materials and Methods

(a) State if IRB has been obtained for all datasets.

(b) Describe the subject groups' demographics.

(c) Have the two studies been conducted simultaneously or sequentially?

(d) State how the cortisol levels were evaluated against the acute and chronic stress standards to highlight the claims.

(e) Subchapter 2.1.1.

-       Use the terminology in full first,

-       State if the three states were sequential or not and describe the psychological questionnaire used for validation,

-       Explain why phases one and three were non-stress/baseline conditions,

-       Briefly explain whether the study used a baseline for chronic stress (the cortisol levels) and spiked with acute episodes (pics of cortisol levels for a predetermined time),

-       Which was the control group for this dataset (specify the benchmarking)?

(f) Subchapter 2.1.2.

-       Briefly explain DASS-21 to connect with labelling and selection based on the scores >7 (the scale as stress interpretation tool).

-       Which was the control group for this dataset (specify the benchmarking)?

-    Were the subjects in the UWS group also enrolled in the MMSD group?

(g) Subchapter 2.2.

-       editing "[21]…."

-       "Table" instead of "table."

-       is “Algorithm 1” a Table?

-       Briefly explain why HRV has been chosen as a stress parameter.

-       Briefly explain the RR and the values.

-       Provide sources of errors and bias that can affect the signals linked to the pre-processing steps.

Results and Discussions

(a) Please integrate the results with examples from the scientific literature to highlight the benefits of the proposed model and its potential implementation.

(b) Compare the test's sensitivity with other models and highlight the novelty of the proposed model.

(c) Address the challenges faced during the study to briefly explain their impact on the results or further implementation and how they have been resolved.

(d) Describe the claims and discuss them concerning the bigger picture: sensitive stress diagnostic cum monitoring model for better generalisation capability.

(e) Discuss how data analytics may contribute to stress management and possibly to medical conditions' prognosis if the prediction is applied further.

Conclusions

Rephrase to help readers understand the workflow (the motivation, the hypothesis, the claims, the novelty, the methods, the results compared with other methods, the challenges faced and solved, and the future results' implementation) from IoT and POCT perspectives. Present the clear big-picture and how the results fit it.

Thank you, and best regards.

Author Response

(The authors gave the same response as above.)

Round 2

Reviewer 1 Report

The authors have made significant changes.

However, first the authors need to clarify the AUC ROC in eq 5. The calculation of AUC is based a summation of several TPRs and FPRs considering several threshold values. Not only a single TPR and FPR. Instead, if you use a Python module to calculate it, please just refer the function in that module.

Another minor issue is about the comparative study table. Please put this table in discussion chapter and please compare your result with them. You can have a discussion here justifying your result and previous results.

- Page 6 "...listed in the table 2" please use capital letter for the given tables in paragraph. 

- Please check the indent of the paragraphs 

Author Response

Thank you once again for your very interesting comments and corrections.

Reviewer 2 Report

Many thanks to the authors for reviewing the manuscript. 

Consider minor editing and adding some info which may increase the readers' understanding of the results and impact of the proposed research.  

1. unless you propose a subtitle 1.2., with different content needed to highlight other concepts and specific approaches for a better understanding of the topic and claims, is it ok to remove subtitle 1.1 and have one body of text as the Introduction?

2. "The RR interval, or "normal-to-normal" (NN) interval, is the time interval between consecutive heartbeats." Please provide the physiological values/range of the RR for the readers to compare, understand the changes under stress and link to the data provided. 

3. "... to interpolation than oth;lk kuyyuiers [26]."

4. about DASS-21: for the readers who may wish to have a clear picture about the stress scale and the related resources, will it be ok to provide info such as: "the stress level has been evaluated on a scale from ... to ... " and "the stress scale was described in ... [citation] "; then the readers can appreciate the tool's efficiency and validity. The authors may add to the Discussions to include the stress levels according to the scale, e.g., low-level: from ... to ...; medium-level: from ... to ...; high-level: from ... to ... .

5. all the explanations regarding the procedure and background and provided in the Cover letter are needed in the text to highlight the authors' scientific decisions. e.g. :...Because this study was done in real-life ... the generalizability was interesting." 

Once again thank you, and best regards!

Author Response

(The authors gave the same response as above.)
